# Learning to Dispatch for Job Shop Scheduling via Deep Reinforcement Learning

**Cong Zhang[1,*], Wen Song[2,*], Zhiguang Cao[3,†], Jie Zhang[1], Puay Siew Tan[4], and Chi Xu[4]**

[1]Nanyang Technological University
[2]Institute of Marine Science and Technology, Shandong University, China
[3]National University of Singapore
[4]Singapore Institute of Manufacturing Technology, A*STAR
cong030@e.ntu.edu.sg, wensong@email.sdu.edu.cn, zhiguangcao@outlook.com
zhangj@ntu.edu.sg, {pstan, cxu}@simtech.a-star.edu.sg

## Abstract

Priority dispatching rule (PDR) is widely used for solving real-world Job-shop scheduling problem (JSSP). However, the design of effective PDRs is a tedious task, requiring a myriad of specialized knowledge and often delivering limited performance. In this paper, we propose to automatically learn PDRs via an end-to-end deep reinforcement learning agent. We exploit the disjunctive graph representation of JSSP, and propose a Graph Neural Network based scheme to embed the states encountered during solving. The resulting policy network is size-agnostic, effectively enabling generalization on large-scale instances. Experiments show that the agent can learn high-quality PDRs from scratch with elementary raw features, and demonstrates strong performance against the best existing PDRs. The learned policies also perform well on much larger instances that are unseen in training.

## 1 Introduction

Job-shop scheduling problem (JSSP) is a well-known combinatorial optimization problem in computer science and operations research, and is ubiquitous in many industries such as manufacturing and transportation [1, 2]. In JSSP, a number of *jobs* with predefined processing constraints (e.g. the operations are processed in order by their eligible machines) are assigned to a set of heterogeneous *machines*, to achieve the desired objective such as minimizing the makespan, flowtime, or tardiness. Due to its NP-hardness, finding exact solutions to JSSP is often impractical [3, 4], while efficiency in practice usually relies on heuristics [5, 6] or approximate methods [7].

Priority dispatching rule (PDR) [6] is a heuristic method that is widely used in real-world scheduling systems. Compared with complicated optimization methods such as mathematical programming and metaheuristics, PDR is computationally fast, intuitive and easy to implement, and naturally capable of handling uncertainties that are ubiquitous in practice [8]. Motivated by these advantages, a large number of PDRs for JSSP have been proposed in the literature [9]. However, it is commonly accepted that designing an effective PDR is very costly and time-consuming, requiring substantial domain knowledge and trial-and-error especially for complex JSSP. Moreover, performance of a PDR often varies drastically on different instances [10]. Therefore, a natural question to ask is: can we *automate* the process of designing PDR, such that it performs well on a class of JSSP instances sharing common characteristics? A number of recent works on learning algorithms for other types of combinatorial optimization problems (COPs) (see [11] for a survey) show that deep reinforcement

---

learning (DRL) could be an ideal technique for this purpose. However, for complex scheduling problems such as JSSP which differs structurally from other COPs and received much less attention, existing methods cannot apply [11], and it remains challenging to design effective representation and learning mechanism.

In this paper, we propose a novel DRL based method to automatically learn strong and robust PDRs for solving JSSP. Specifically, we first present a Markov Decision Process (MDP) formulation of PDR based scheduling, where the states are captured by leveraging the *disjunctive graph* representation of JSSP. Such representation effectively integrates the operation dependencies and machine status, and provides critical information for scheduling decisions. Then, we propose a Graph Neural Network (GNN) based scheme with an efficient computation strategy to encode the nodes in the disjunctive graphs to fixed dimensional embeddings. Based on this scheme, we design a size-agnostic policy network that can process JSSP instances with arbitrary size, which effectively enables training on small-sized instances and generalizing to large-scale ones. We train the network using a policy gradient algorithm to obtain high-quality PDRs, without the need of supervision. Extensive experiments on generated instances and standard benchmarks show that, the PDRs trained by our policy significantly outperform existing manually designed ones, and generalize reasonably well to instances that are much larger than those used in training.

## 2   Related Work

Lately, the idea of applying deep (reinforcement) learning as an end-to-end solution to combinatorial optimization problems has been widely explored. Most of them focus on solving routing problems (e.g. travelling salesman problem) [12, 13, 14, 15, 16, 17], graph optimization problems [18, 19], and the satisfiability problem (SAT) [20, 21, 22]. In contrast, scheduling problems which have numerous real-world applications, are relatively unexplored, especially for JSSP.

Several existing works study simple job scheduling problems, in which jobs as elementary tasks without internal operation dependencies that are essential to JSSP. In [23], a DRL agent is proposed to learn job scheduling policies for a compute cluster. A 2-D image based state representation scheme is used to capture the status of resources and jobs. In [24], DRL is employed to learn local search heuristics for solving a similar problem, where the states are represented by a Directed Acyclic Graph (DAG) describing the temporal relations among jobs in the corresponding schedule. A major limitation in these works is that, the state representation is hard-bounded by some factors (e.g. look ahead horizon, size of job queue or slot), and is not scalable to arbitrary numbers of jobs and machines (resources). This limitation is partially alleviated in [25], which also employs an image based representation but with a transfer learning method to reconstruct the trained policies on problems with different sizes. Nevertheless, policy transfer is still relatively costly and inconvenient. In contrast, our method is fully size-agnostic and the trained policy can be directly applied to solve larger problems without the need of transfer.

In [26], a DRL method is proposed for task scheduling in a cloud computing environment. GNN is used to extract embedding of each task represented as a DAG, and the policy network can scale to arbitrary number of tasks. However, the underlying problem is not JSSP and the task DAG only describes the required temporal dependencies among its subtasks. The resource information is encoded as node features, hence the number of resources is hard bounded. In contrast, our GNN performs embedding on the disjunctive graph with directed disjunctive arcs reflecting processing order on each machine, and is size-agnostic in terms of both jobs and machines. Moreover, the topology of task DAGs in [26] is static, while the disjunctive graph in our setting is dynamically evolving and highly correlated with the decisions made at each step. Similar to [26], a RL method combined with GNNs is proposed to accelerating computation in distributed system [27]. Other examples of applying GNNs to solve real life scheduling problem includes [28], where they adopted an imitation learning algorithm for solving robotic scheduling problems in manufacturing.

Research on standard JSSP is rather sparse. In [29], an imitation learning method is proposed to learn dispatching rules for JSSP, where optimal solutions to the training instances are labelled using a MIP solver. However, due to the complexity of JSSP, finding enough optimal solutions to large-scale instances for training is impractical, and only instances up to $10 \; jobs \times 10 \; machines$ are considered, which significantly limits the applicability. In [30], a Deep Q Network [31] based method is proposed to solve JSSP, which learns to select PDR for each machine from a pool of candidates. Though the

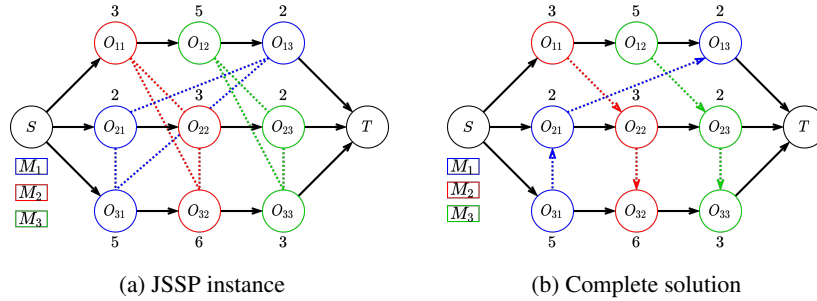

| (a) JSSP instance | (b) Complete solution |

Figure 1: **Disjunctive graph representation.** (a) represents a $3 \times 3$ JSSP instance. Black arrows are conjunctive arcs, and dotted lines are disjunctive arcs grouped into machine cliques with different colors. (b) is a complete solution, where all disjunctive arcs are assigned with directions.

methods in [29, 30] show better performance than traditional PDRs, they are not end-to-end, and rely on manual features to describe scheduling states, ignoring the underlying graph structure of JSSP. In contrast, our method can extract informative knowledge from low-level raw features based on the disjunctive graph, and directly makes decisions without the need of pre-defined candidate PDRs.

## 3 Preliminaries

**Job-Shop Scheduling Problem.** A standard JSSP instance consists of a set of jobs $\mathcal{J}$ and a set of machines $\mathcal{M}$. Each job $J_i \in \mathcal{J}$ must go through $n_i$ machines in $\mathcal{M}$ in a specific order denoted as $O_{i1} \to ... \to O_{in_i}$, where each element $O_{ij}$ ($1 \le j \le n_i$) is called an operation of $J_i$ with a processing time $p_{ij} \in \mathbb{N}$. The binary relation $\to$ also refers to precedence constraint. Each machine can only process one job at a time, and preemption is not allowed. To solve a JSSP instance, we need to find a *schedule*, i.e. starting time $S_{ij}$ for each operation $O_{ij}$, such that the *makespan* denoted as $C_{\max} = \max_{i,j}\{C_{ij} = S_{ij} + p_{ij}\}$ is minimized and all the constraints are satisfied. The size of a JSSP instance is denoted as $|\mathcal{J}| \times |\mathcal{M}|$.

**Disjunctive graph.** It is well-known that a JSSP instance can also be represented as a disjunctive graph [32]. Let $\mathcal{O} = \{O_{ij}|\forall i, j\} \cup \{S, T\}$ be the set of all operations, where $S$ and $T$ are dummy ones denoting the start and terminal with zero processing time. Then a disjunctive graph $G = (\mathcal{O}, \mathcal{C}, \mathcal{D})$ is a mixed graph with $\mathcal{O}$ being its vertex set. In particular, $\mathcal{C}$ is a set of directed arcs (conjunctions) representing the precedence constraints between operations of the same job; and $\mathcal{D}$ is a set of undirected arcs (disjunctions), each of which connects a pair of operations requiring the same machine for processing. Consequently, finding a solution to a JSSP instance is equivalent to fixing the direction of each disjunction, such that the resulting graph is a DAG [33]. An example of disjunctive graph for a JSSP instance and its solution are depicted in Figure 1(a) and (b), respectively.

## 4 Method

In this section, we present the rationale of our method. We first formulate Markov Decision Process model of PDR based scheduling. Then, we design an efficient method to represent the scheduling policy based on Graph Neural Network, followed by the introduction of training algorithm.

### 4.1 Markov Decision Process Formulation

The PDR based method solves a JSSP instance using $|\mathcal{O}|$ steps of consecutive decisions. At each step, a set of eligible operations (i.e. those whose precedent operation has been scheduled) are identified first. Then, a specific PDR is applied to compute a priority index for each eligible operation, and the one with the highest priority is selected for scheduling (or dispatching). However, solely deciding which operation to dispatch is not sufficient, as we also need to choose a suitable start time for it. In order to build tight schedules, it is sensible to place the operation as early as possible on the corresponding machine [29]. Once all operations are dispatched, a complete schedule is generated.

Traditional manually designed PDRs compute the priority index based on the operation features. For example, the widely used Shortest Processing Time (SPT) rule selects from a set of operations the one with the smallest $p_{ij}$. In this paper, we employ DRL to automatically generate high-quality PDRs. As mentioned above, solving a JSSP instance can be viewed as a task of determining the direction of each disjunction. Therefore, we consider the dispatching decisions made by PDRs as actions of changing the disjunctive graph, and formulate the underlying MDP model as follows.

**State.** The state $s_t$ at decision step $t$ is a disjunctive graph $G(t) = (\mathcal{O}, \mathcal{C} \cup \mathcal{D}_u(t), \mathcal{D}(t))$ representing the current status of solution, where $\mathcal{D}_u(t) \subseteq \mathcal{D}$ contains all the (directed) disjunctive arcs that have been assigned a direction till $t$, and $\mathcal{D}(t) \subseteq \mathcal{D}$ includes the remaining ones. The initial state $s_0$ is the disjunctive graph representing the original JSSP instance, and the terminal state $s_T$ is a complete solution where $\mathcal{D}(T) = \emptyset$, i.e. all disjunctive arcs have been assigned a direction. For each node $O \in \mathcal{O}$, we record two features: 1) a binary indicator $I(O, s_t)$ which equals to 1 only if $O$ is scheduled in $s_t$, and 2) an integer $C_{LB}(O, s_t)$ which is the lower bound of the estimated time of completion (ETC) of $O$ in $s_t$. Note that for the scheduled operation, this lower bound is exactly its completion time. For the unscheduled operation $O_{ij}$ of job $J_i$, we recursively calculate this lower bound as $C_{LB}(O_{ij}, s_t) = C_{LB}(O_{i,j-1}, s_t) + p_{ij}$ by only considering the precedence constraints from its predecessor, i.e. $O_{i,j-1} \rightarrow O_{ij}$, and $C_{LB}(O_{ij}, s_t) = r_i + p_{ij}$ if $O_{ij}$ is the first operation of $J_i$ where $r_i$ is the release time of $J_i$.

**Action.** An action $a_t \in A_t$ is an eligible operation at decision step $t$. Given that each job can only have at most one operation ready at $t$, the maximum size of action space is $|\mathcal{J}|$, which depends on the instance being solved. During solving, $|A_t|$ becomes smaller as more jobs are completed.

**State transition.** Once PDR determines an operation $a_t$ to dispatch next, we first find the earliest feasible time period to allocate $a_t$ on the required machine. Then, we update the directions of the disjunctive arcs of that machine based on the current temporal relations, and engenders a new disjunctive graph as the new state $s_{t+1}$. An example is given in Figure 2, where action $a_4 = O_{32}$ is chosen at state $s_4$ from action space $\{O_{12}, O_{23}, O_{32}\}$. On the required machine $M_2$, we find that $O_{32}$ can be allocated in the time period before the already scheduled $O_{22}$, therefore the direction of the disjunctive arc between $O_{22}$ and $O_{32}$ is determined as $O_{32} \rightarrow O_{22}$, as shown in the new state $s_5$. Note, the starting time of $O_{22}$ is changed from 7 to 11 since $O_{32}$ is scheduled before $O_{22}$.

**Reward.** The goal is to learn to dispatch step by step such that the makespan is minimized. To this end, we design the reward function $R(s_t, a_t)$ as the quality difference between the partial solutions corresponding to the two states $s_t$ and $s_{t+1}$, i.e. $R(a_t, s_t) = H(s_t) - H(s_{t+1})$, where $H(\cdot)$ is the quality measure. Here we define it as the lower bound of the makespan $C_{\max}$, computed as $H(s_t) = \max_{i,j}\{C_{LB}(O_{ij}, s_t)\}$. For the terminal state $s_{|\mathcal{O}|}$, clearly we have $H(s_{|\mathcal{O}|}) = C_{\max}$ since all operations are scheduled. Hence when the discount factor $\gamma = 1$, the cumulative reward is $\sum_{t=0}^{|\mathcal{O}|} R(a_t, s_t) = H(s_0) - C_{\max}$. Given that $H(s_0)$ is constant, maximizing the cumulative reward coincides with minimizing the makespan.

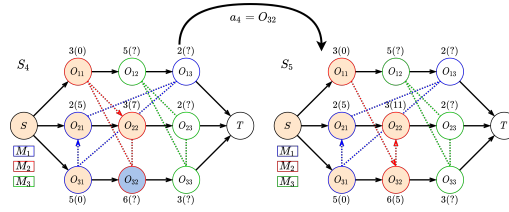

Figure 2: **Example of state transition.** Orange nodes are operations already scheduled, and the grey node is operation selected to be scheduled at current state. Integers in bracket are starting time of scheduled operations, where unscheduled operations have unknown starting time (denoted as ?).

**Policy.** For state $s_t$, a stochastic policy $\pi(a_t|s_t)$ outputs a distribution over the actions in $A_t$. If traditional PDRs are employed as policy, then the distribution is one-hot, and the action with the highest priority has probability 1.

## 4.2 Parameterizing the Policy

The disjunctive graph in the above MDP formulation provides a holistic view of the scheduling states that comprehensively contains numerical and structural information such as operation processing time, precedence constraints, and processing order on each machine. By extracting all the state information embedded in disjunctive graphs, effective dispatching is viable. This motivates us to

parameterize the stochastic policy $\pi(a_t|s_t)$ as a graph neural network with trainable parameter $\theta$, i.e. $\pi_\theta(a_t|s_t)$, which enables learning strong dispatching rules and size-agnostic generalization.

**Graph embedding.** Graph Neural Networks (GNN) [34] are a family of deep neural networks that can learn representation of graph-structured (non-euclidean) data, which has many applications in real life [35, 36]. It extracts feature embedding of each node in an iterative and non-linear fashion.In this paper, we adopt the Graph Isomorphism Network (GIN) [37], which is a recent GNN variant and proved to have strong discriminative power. Particularly, given a graph $\mathcal{G} = (V, E)$, GIN performs $K$ iterations of updates to compute a $p$-dimensional embeddings for each node $v \in V$, and the update at iteration $k$ is expressed as follows,

$$h_v^{(k)} = MLP_{\theta_k}^{(k)} \left( \left( 1 + \epsilon^{(k)} \right) \cdot h_v^{(k-1)} + \sum_{u \in \mathcal{N}(v)} h_u^{(k-1)} \right), \tag{1}$$

where $h_v^{(k)}$ is the representation of node $v$ at iteration $k$ and $h_v^{(0)}$ refers to its raw features for input, $MLP_{\theta_k}^{(k)}$ is a Multi-Layer Perceptron (MLP) with parameter $\theta_k$ for iteration $k$ followed by batch normalization [38], $\epsilon$ is an arbitrary number that can be learned, and $\mathcal{N}(v)$ is the neighbourhood of $v$. After $K$ iterations of updates, a global representation for the entire graph can be obtained using a pooling function $L$ that takes as input the embeddings of all nodes and output a $p$-dimensional vector $h_\mathcal{G} \in \mathbb{R}^p$ for $\mathcal{G}$. Here we use average pooling, i.e. $h_\mathcal{G} = L(\{h_v^K : v \in V\}) = 1/|V| \sum_{v \in V} h_v^K$.

GIN is originally proposed for undirected graphs in [37]. However, in our case, the disjunctive graph $G(t)$ correlative to each state $s_t$ is a mixed graph with directed arcs, which describe critical characteristics such as the precedence constraints and operation sequences on machines. Therefore, we need to generalize GIN to support disjunctive graphs. A natural and straightforward strategy for this is to replace each undirected arc in $G(t)$ by two directed ones connecting the same nodes with opposite directions, resulting in a fully directed graph denoted as $G_D(t)$ [39, 40]. Then, the neighbourhood of node $v$ in Eq. (1) can be defined as $\mathcal{N}(v) = \{u|(u,v) \in E(G_D(t))\}$, where $E(\cdot)$ is the arc set of a graph, i.e. $\mathcal{N}(v)$ contains all incoming neighbours of $v$. In this way, GIN is able to operate on $G_D(t)$. An illustration is given in Figure 3(a), which is the directed version of the state $s_4$ in Figure 2. The transition to $s_5$ can be naturally achieved by removing directed arcs $(O_{32}, O_{11})$ and $(O_{22}, O_{32})$ in Figure 3(b) since the direction of the corresponding disjunctive arcs should be $O_{11} \to O_{32}$ and $O_{32} \to O_{22}$.

However, a major limitation of the above "removing-arc" strategy is that, it maintains two directed arcs for each disjunctive arc, making $G_D(t)$ too dense to be efficiently processed by GIN. This is more severe for the initial states, where for each machine the operations requiring it forms a clique that is fully connected with $|\mathcal{J}|(|\mathcal{J}| - 1)/2$ arcs in $G(t)$. To resolve this issue, we propose to "add" arcs rather than removing them, where the undirected disjunctive arcs are neglected. More specifically, we use $\tilde{G}_D(t) = (\mathcal{O}, \mathcal{C} \cup \mathcal{D}_u(t))$ as an approximation of $G_D(t)$. Along with the scheduling process, $\mathcal{D}_u(t)$ becomes larger since more directed arcs will be added to it. An illustration of this "adding-arc" strategy is shown in Figure 3(c). Clearly, this strategy leads to much sparser graphs for state representation. Finally, we define the raw features for each node $O \in \mathcal{O}$ at $s_t$ as a 2-dimensional vector $h_O^{(0)}(s_t) = (I(O, s_t), C_{LB}(O, s_t))$, and denote the node and graph embedding obtained after $K$ iterations as $h_O^{(K)}(s_t)$ and $h_\mathcal{G}(s_t)$, respectively.

**Action selection.** To select an action $a_t$ at $s_t$, we further process the extracted graph embeddings $h_O^{(K)}$ with an action selection network. In doing so, we expect to produce a probability distribution over action space from which $a_t$ can be sampled. Specifically, we first adopt an MLP to obtain a scalar sore $scr(a_t) = MLP_{\theta_\pi}([h_{a_t}^{(K)}, h_\mathcal{G}(s_t)])$ for each $a_t$, where $[,]$ means concatenation. Then, a softmax function is applied to output a distribution $P(a_t)$ over computed scores. We sample actions based on $P(a_t)$ for training. During testing, we greedily pick $a_t$ with the maximum probability.

*Remark.* Our design of policy network has several advantages. First, unlike previous works [23, 24, 25], it is not hard bounded by the instance size ($|\mathcal{J}|$ and $|\mathcal{M}|$), since all parameters are shared across all nodes in the graph. This size-agnostic property effectively enables generalization to instances of different sizes without re-training or knowledge transferring. Second, it can potentially deal with more complex environments with dynamics and uncertainty such as job arriving on-the-fly and random machine breakdown, by adding or removing certain nodes and/or arcs from the disjunctive graphs. Finally, our model could be extended to other shop scheduling problems (e.g. flow-shop and

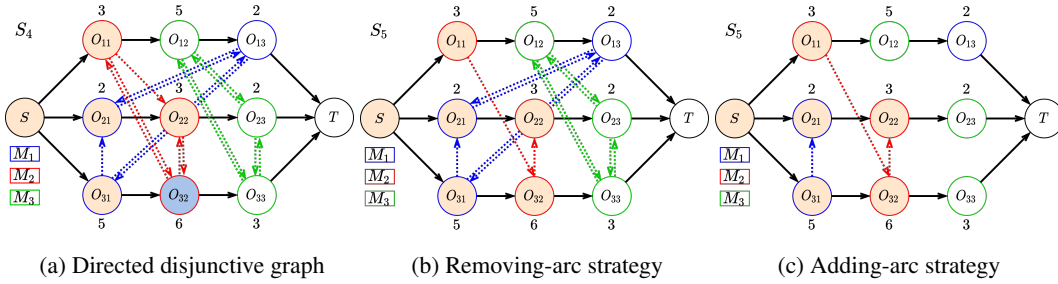

(a) Directed disjunctive graph      (b) Removing-arc strategy      (c) Adding-arc strategy

Figure 3: **Fully directed disjunctive graph, removing-arc strategy, and adding-arc strategy.** (a) is a fully directed disjunctive graph by replacing each undirected disjunctive arc with two opposite directed arcs. (b) shows the removing-arc strategy. The directed arc conflicting with the current decision is removed from the graph, i.e. arcs $(O_{32}, O_{11})$ and $(O_{22}, O_{32})$. (c) shows the adding-arc strategy. The directed arc following the current decision is added to the graph, i.e. arcs $(O_{11}, O_{32})$ and $(O_{32}, O_{22})$.

open-shop) since they can be represented by disjunctive graphs [41, 42]. While the first advantage will be demonstrated in the experiments, we plan to further explore the latter two in the future.

### 4.3 Learning Algorithm

We train the policy network using Proximal Policy Optimization (PPO) [43], which is an actor-critic algorithm. The actor refers to the policy network $\pi_\theta$ described above. The critic $v_\phi$ shares the same GIN network with the actor, and uses an MLP $MLP_{\theta_v}$ that takes input of $h_{\mathcal{G}}(s_t)$ to output a scalar to estimate the cumulative rewards at $s_t$. To boost learning, we follow the principle of PPO and generate $N$ independent trajectories, and update network parameters w.r.t cumulative gradient of $N$ estimates. Details of the training algorithm are provided in the Supplementary Material.

## 5 Experiment

We present the experimental results in this section. The evaluations are performed both on generated instances and public JSSP benchmarks.

**Datasets.** We evaluate our method on instances of various sizes. Specifically, $6 \times 6$, $10 \times 10$, $15 \times 15$, $20 \times 20$, and $30 \times 20$ instances are generated following the well-known Taillard's method [44] for training and testing. Furthermore, we demonstrate strong generalization of our method by directly testing on much larger instances with sizes $50 \times 20$ and $100 \times 20$ generated also following [44]. We also perform experiments on well-known public JSSP benchmarks, including Taillard's instances [45] generated following [44] and the DMU instances [46]. The range of operation processing times in DMU instances doubles that of Taillard's ones.

**Models and configurations.** We use fixed hyperparameters for training. For each problem size, we train the policy network for 10000 iterations, each of which contains 4 independent trajectories (i.e. instances). The model is validated on 100 instances generated on-the-fly and fixed during training. All raw features are normalized to the same scale. For the GIN layers (Eq. (1)) shared by $\pi$ and $v$, we set the number of iterations $K = 2$. We set $\epsilon$ to 0 following [37]. Each $MLP_{\theta_k}^{(k)}$ in the GIN layer has 2 hidden layers with hidden dimension 64. The action selection network $MLP_{\theta_\pi}$ and state value prediction network $MLP_{\theta_v}$ both have 2 hidden layers with hidden dimension 32. For PPO, we set the epochs of updating network to 1, the clipping parameter $\epsilon_{PPO}$ to 0.2, and the coefficient for policy loss, value function, and entropy to 2, 1, and 0.01, respectively. For training, we set the discount factor $\gamma$ to 1, and use the Adam optimizer with constant learning rate $lr = 2 \times 10^{-5}$. Other parameters follow the default settings in PyTorch [47]. The hardware we use is a machine with Intel Core i9-10940X CPU and a single Nvidia GeForce 2080Ti GPU. Our code is available.[1]

| Size | | SPT | MWKR | FDD/MWKR | MOPNR | Ours | Opt. Rate(%) |
|---|---|---|---|---|---|---|---|
| $6 \times 6$ | Obj. | 691.95 | 656.95 | 604.64 | 630.19 | **574.09** | |
| | Gap | 42.0% | 34.6% | 24.0% | 29.2% | **17.7%** | 100% |
| | Time(s) | **0.012** | **0.012** | **0.012** | **0.012** | 0.061 | |
| $10 \times 10$ | Obj. | 1210.98 | 1151.41 | 1102.95 | 1101.08 | **988.58** | |
| | Gap | 50.0% | 42.6% | 36.6% | 36.5% | **22.3%** | 100% |
| | Time(s) | **0.037** | 0.039 | 0.039 | **0.037** | 0.176 | |
| $15 \times 15$ | Obj. | 1890.91 | 1812.13 | 1722.73 | 1693.33 | **1504.79** | |
| | Gap | 59.2% | 52.6% | 45.1% | 42.6% | **26.7%** | 99% |
| | Time(s) | 0.113 | 0.116 | 0.117 | **0.112** | 0.435 | |
| $20 \times 20$ | Obj. | 2519.8 | 2469.19 | 2328.15 | 2263.68 | **2007.76** | |
| | Gap | 62.0% | 58.6% | 49.6% | 45.5% | **29.0%** | 4% |
| | Time(s) | 0.306 | 0.312 | 0.312 | **0.305** | 0.932 | |
| $30 \times 20$ | Obj. | 3208.69 | 3080.11 | 2883.88 | 2809.62 | **2508.27** | |
| | Gap | 65.3% | 58.7% | 48.6% | 44.7% | **29.2%** | 12% |
| | Time(s) | 0.721 | 0.731 | 0.731 | **0.720** | 1.804 | |

Table 1: **Results on instances of small and medium sizes.** "Opt. Rate": rate of instances for which OR-Tools returns optimal solution.

**Baselines.** There are hundreds of PDRs proposed for JSSP in the literature with various performance and we can not compare with them exhaustively. Therefore, we select four traditional PDRs based on their performance reported in [9], including *Shortest Processing Time* (SPT), *Most Work Remaining* (MWKR), *Most Operations Remaining* (MOPNR), and *minimum ratio of Flow Due Date to Most Work Remaining* (FDD/MWKR). SPT is one of the most widely used PDRs in research and industry, while the other three are top-performing PDRs on Taillard's benchmark as reported in [9]. Specifically, FDD/MWKR is newly developed in [9]. All baselines are implemented in Python, and the details are introduced in the Supplementary Material. For the generated instances, solutions of all methods are benchmarked with those obtained by Google OR-Tools [48], a mature and widely used exact solver based on constraint programming, with time limit of 3600 seconds for each instance. For the public benchmarks, we use the best-known solutions from the literature.[2]

## 5.1 Results on Generated Instances

We first perform training and testing on the generated instances of small to medium sizes ($6 \times 6$, $10 \times 10$, $15 \times 15$, $20 \times 20$, $30 \times 20$). For each size, we generate 100 instances randomly and report the average objective, gap to the OR-Tools solutions, and computational time of our method and baselines. The results are summarized in Table 1. We can observe from this table that the PDR learned by our method consistently outperforms all baseline PDRs by a large margin regarding all instance sizes. Performance of baseline PDRs deteriorates quickly with the increase of instance size, whereas PDR learned by our method performs stable and relatively well especially on larger instances. In terms of computational efficiency, though the inference time of our method is relatively longer than the traditional PDRs, it is still quite acceptable especially considering the significant performance boost. Compared with OR-Tools which takes 3600s on the vast majority of $20 \times 20$ and $30 \times 20$ instances, our method is much more efficient. Based on the above observations, we can conclude that our method is able to train high-quality PDRs from scratch, without the need of supervision.

Next, we evaluate the performance of our policy in terms of generalizing to large instances. More specifically, we directly use the policies trained on $20 \times 20$ and $30 \times 20$ instances to solve $50 \times 20$ and $100 \times 20$ instances. The results are summarized in Table 2, where the result for each instance size is averaged over 100 random instances. As shown in this table, both our policies trained on much smaller instances perform reasonably well on these large sized ones, and deliver solutions that are much better than those of the traditional PDRs. This observation shows that our method is able to extract knowledge from small sized instances that is also useful in solving large-scale ones, which is a desirable property for practical applications. Meanwhile, our method is computationally efficient and can provide high-quality solution for the largest instance within 30s. We can also observe that

| Size | | SPT | MWKR | FDD/MWKR | MOPNR | Ours (20 × 20) | Ours (30 × 20) | Opt. Rate |
|---|---|---|---|---|---|---|---|---|
| 50 × 20 | Obj. | 4469.8 | 4273.08 | 3993.45 | 3859.14 | 3581.5 | **3522.5** | 48% |
| | Gap | 54.9% | 48.1% | 38.4% | 33.7% | 24.1% | **22.1%** | |
| | Time(s) | **2.504** | 2.523 | 2.524 | **2.504** | 4.917 | 4.872 | |
| 100 × 20 | Obj. | 7516.12 | 7069.72 | 6658.17 | 6385.32 | 6175.01 | **6088.68** | 2% |
| | Gap | 35.1% | 27.0% | 19.6% | 14.7% | 10.9% | **9.4%** | |
| | Time(s) | 16.661 | 16.694 | 16.723 | **16.625** | 27.869 | 28.616 | |

Table 2: **Generalization results on large-sized instances.** "Opt. Rate": rate of instances that OR-Tools returns optimal solution.

| Size | | SPT | MWKR | FDD/MWKR | MOPNR | Ours | Ours (20 × 20) | Ours (30 × 20) | Opt. Rate |
|---|---|---|---|---|---|---|---|---|---|
| Ta 15 × 15 | Obj. | 1902.6 | 1927.5 | 1808.6 | 1782.3 | **1547.4** | . | . | 100% |
| | Gap | 54.8% | 56.7% | 47.1% | 45.0% | **26.0%** | . | . | |
| | Time(s) | **0.111** | 0.115 | 0.117 | **0.111** | 0.447 | . | . | |
| Ta 20 × 15 | Obj. | 2253.6 | 2190.7 | 2054 | 2015.8 | **1774.7** | . | . | 90% |
| | Gap | 65.2% | 60.7% | 50.6% | 47.7% | **30.0%** | . | . | |
| | Time(s) | **0.178** | 0.183 | 0.183 | **0.178** | 0.624 | . | . | |
| Ta 20 × 20 | Obj. | 2655.8 | 2518.6 | 2387.2 | 2309.9 | **2128.1** | . | . | 30% |
| | Gap | 64.2% | 55.7% | 47.6% | 42.8% | **31.6%** | . | . | |
| | Time(s) | 0.305 | 0.311 | 0.311 | **0.304** | 0.937 | . | . | |
| Ta 30 × 15 | Obj. | 2888.4 | 2728 | 2590.8 | 2601.3 | **2378.8** | . | . | 70% |
| | Gap | 61.6% | 52.6% | 45.0% | 45.6% | **33.0%** | . | . | |
| | Time(s) | **0.383** | 0.390 | 0.392 | **0.383** | 1.114 | . | . | |
| Ta 30 × 20 | Obj. | 3234.7 | 3193.3 | 3045 | 2888.1 | **2603.9** | . | . | 0% |
| | Gap | 66.0% | 63.9% | 56.3% | 48.2% | **33.6%** | . | . | |
| | Time(s) | 0.722 | 0.730 | 0.731 | **0.720** | 1.799 | . | . | |
| Ta 50 × 15 | Obj. | 4194.7 | 3907.8 | 3736.3 | 3608 | . | 3430.2 | **3393.8** | 100% |
| | Gap | 51.4% | 40.9% | 34.8% | 30.1% | . | 23.7% | **22.4%** | |
| | Time(s) | **1.208** | 1.221 | 1.226 | 1.209 | . | 2.700 | 2.696 | |
| Ta 50 × 20 | Obj. | 4532.2 | 4375.1 | 4022.1 | 3920 | . | 3611.8 | **3593.9** | 100% |
| | Gap | 59.5% | 53.9% | 41.5% | 37.9% | . | 27.0% | **26.5%** | |
| | Time(s) | **2.507** | 2.527 | 2.523 | 2.508 | . | 4.856 | 4.883 | |
| Ta 100 × 20 | Obj. | 7564.6 | 7128.8 | 6620.7 | 6452.3 | . | 6255 | **6097.6** | 100% |
| | Gap | 41.0% | 32.9% | 23.4% | 20.2% | . | 16.6% | **13.6%** | |
| | Time(s) | 16.652 | 16.686 | 16.745 | **16.647** | . | 28.239 | 28.328 | |

Table 3: **Results on Taillard's benchmark.** "Opt. Rate": rate of instances with optimal solution.

our 20 × 20 policy performs only slightly worse than the 30 × 20 one, indicating a relatively robust generalization performance.

## 5.2 Results on Public Benchmarks

We first perform experiments on the 80 Taillard's instances, which can be classified into 8 groups according to their sizes, each with 10 instances. We train a policy for each of the 5 groups up to 30 × 20, while the remaining 3 groups are used for the generalization test. The results for each group are summarized in Table 3, while the detailed results for each instance can be found in the Supplementary Material. Note that the gaps are calculated using the best solutions in the literature. As shown in this table, the PDRs learned by our method still maintain good performance on the Taillard's benchmark and produce solutions significantly better than baseline PDRs, both when evaluating on the same size and when generalizing to larger size. It is interesting to see that all methods show smaller gaps on large-sized instances (50 × 15, 50 × 20 and 100 × 20), which is probably because instances with larger $|\mathcal{J}|/|\mathcal{M}|$ tend to be easier to solve as noticed in [44]. But still, these instances could be hard for exact solvers due to the NP-hardness of JSSP, as OR-Tools only solves 2% of 100 × 20 instances optimally within the 3600s time limit, as shown in Table 2.

| Size | | SPT | MWKR | FDD/MWKR | MOPNR | Ours (30 × 20) | Opt. Rate |
|---|---|---|---|---|---|---|---|
| Dmu 30 × 20 | Obj. | 7036 | 6925 | 6827.3 | 6491.9 | **5967.4** | 10% |
| | Gap | 65.9% | 63.2% | 60.1% | 52.0% | **39.5%** | |
| | Time(s) | **0.709** | 0.718 | 0.721 | **0.709** | 1.805 | |
| Dmu 50 × 15 | Obj. | 8975.4 | 8906 | 9150.2 | 8436.5 | **8179.4** | 50% |
| | Gap | 50.4% | 48.9% | 52.5% | 40.8% | **36.2%** | |
| | Time(s) | **1.195** | 1.208 | 1.220 | 1.197 | 2.694 | |
| Dmu 50 × 20 | Obj. | 10132.8 | 9807 | 9899.6 | 9408 | **8751.6** | 50% |
| | Gap | 62.2% | 56.4% | 57.3% | 49.6% | **38.9%** | |
| | Time(s) | **2.496** | 2.516 | 2.521 | 2.500 | 4.908 | |

Table 4: **Results on DMU benchmark.** "Opt. Rate": rate of instances with optimal solution.

Next, we conduct experiments on the 80 DMU instances, which can also be classified into 8 groups according to their sizes. Here we train a policy for each of the 4 groups up to 30 × 20, with the remaining 4 groups as test sets for generalization. Due to limited space, we only present results of the 30 × 20 policy and its generalization performance on 50 × 15 and 50 × 20 instances in Table 4. We can see that our policy still outperforms baselines on these instances with reasonable time. Complete results on DMU benchmark are given in the Supplementary Material.

# 6 Conclusions and Future Work

In this paper, we present an end-to-end DRL based method to automatically learn high-quality PDRs for solving JSSP. Based on the disjunctive graph representation of JSSP, we propose an MDP formulation of the PDR based scheduling process. Then we design a size-agnostic policy network based on GNN, such that the patterns contained in the graph structure of JSSP can be effectively extracted and reused to solve instances of different sizes. Extensive experiments on generated and public benchmark instances well confirm the superiority of our method to the traditional manually designed PDRs. In the future, we plan to further enhance the performance of our method, and extend it to support other types of shop scheduling problems and complex environments with uncertainties.

## Broader Impact

Some work [49, 50] discussed the design of intelligent production systems by integrating modern AI technology. Our work, which solves a well-known problem that is ubiquitous in real-world production system, i.e. job shop scheduling, is within this scope. The automated end-to-end learning system in this work tries to free human labor from tedious work of designing effective dispatching rules for particular job shop scheduling problem. On the other side, however, this work may have some limitations. First, despite of performance improvement, it sacrifices interpretability due to the unexplainable nature of deep neural networks, whereas traditional dispatching rule based scheduling system is intuitive to human. This issue might make it untrustworthy for some applications, due to the potential risk and uncertainty. Second, highly automated and end-to-end system may conceal some details that are critical but easy to be ignored and bias human's understanding underneath.

## Acknowledgments

We thank Yaoxin Wu, Liang Xin, Xiao Sha, Yue Han, and Rongkai Zhang for fruitful discussions. Cong Zhang would also like to thank Krysia Broda (from Imperial College London) and Fu-lai Chung (from The Hong Kong Polytechnic University), who taught him research skills before he went to NTU. This work was supported by the A*STAR Cyber-Physical Production System (CPPS) – Towards Contextual and Intelligent Response Research Program, under the RIE2020 IAF-PP Grant A19C1a0018, and Model Factory@SIMTech. Wen Song was partially supported by the Young Scholar Future Plan of Shandong University (Grant No. 62420089964188). Zhiguang Cao was partially supported by the National Natural Science Foundation of China (61803104) and Singapore National Research Foundation (NRF-RSS2016004).

## Footnotes

[1]https://github.com/zcajiayin/L2D

[2]The best solutions for Taillard's and DMU instances can be found in http://optimizizer.com/TA.php and http://jobshop.jjvh.nl/, respectively.

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
