[Supplementary Material]

# Learning to Dispatch for Job Shop Scheduling via Deep Reinforcement Learning

**Cong Zhang[1], Wen Song[2], Zhiguang Cao[3], Jie Zhang[1], Puay Siew Tan[4], and Chi Xu[4]**

[1]Nanyang Technological University
[2]Institute of Marine Science and Technology, Shandong University, China
[3]National University of Singapore
[4]Singapore Institute of Manufacturing Technology, A*STAR

## 1   Details of the Training Algorithm

In the paper, we adopt the Proximal Policy Optimization (PPO) algorithm [36] to train our agent. Here we provide details of our algorithm in terms of pseudo code, as shown in Algorithm 1. Similar to the original PPO in [36], we also use $N$ actors, each solves one JSSP instance drawn from a distribution $\mathbb{D}$. The difference to [36] is that, instead of sampling a batch of data, we use all data collected by the $N$ actors to perform update, i.e. line 13, 14, 15, and 19 in the psuedo code.

---

**Algorithm 1:** PPO learning to dispatch

> **Input** : actor network $\pi_\theta$ and behaviour actor network $\pi_{\theta_{old}}$, with trainable parameters $\theta_{old} = \theta$; critic network $v_\phi$ with trainable parameters $\phi$; number of training steps $U$; discounting factor $\gamma$; update epoch $K$; policy loss coefficient $c_p$; value function loss coefficient $c_v$; entropy loss coefficient $c_e$; clipping ratio $\epsilon$.

1  Initialize $\pi_\theta$, $\pi_{\theta_{old}}$, and $v_\phi$ ;
2  **for** $u = 1, 2, ..., U$ **do**
3      Draw $N$ JSSP instances from $\mathbb{D}$;
4      **for** $n = 1, 2, ..., N$ **do**
5          **for** $t = 0, 1, 2, ...$ **do**
6              Sample $a_{n,t}$ based on $\pi_{\theta_{old}}(a_{n,t}|s_{n,t})$;
7              Receive reward $r_{n,t}$ and next state $s_{n,t+1}$;
8              $\hat{A}_{n,t} = \sum_0^t \gamma^t r_{n,t} - v_\phi(s_{n,t}), r_{n,t}(\theta) = \frac{\pi_\theta(a_{n,t}|s_{n,t})}{\pi_{\theta_{old}}(a_{n,t}|s_{n,t})}$;
9              **if** $s_{n,t}$ *is terminal* **then**
10                 break;
11             **end**
12         **end**
13         $L_n^{CLIP}(\theta) = \sum_0^t \min\left(r_{n,t}(\theta)\hat{A}_{n,t}, \text{clip}\left(r_{n,t}(\theta), 1 - \epsilon, 1 + \epsilon\right)\hat{A}_{n,t}\right)$;
14         $L_n^{VF}(\phi) = \sum_0^t \left(v_\phi(s_{n,t}) - \hat{A}_{n,t}\right)^2$;
15         $L_n^{S}(\theta) = \sum_0^t S(\pi_\theta(a_{n,t}|s_{n,t}))$, where $S(\cdot)$ is entropy;
16         Aggregate losses: $L_n(\theta, \phi) = c_p L_n^{CLIP}(\theta) - c_v L_n^{VF}(\phi) + c_e L_n^{S}(\theta)$ ;
17     **end**
18     **for** $k = 1, 2, ..., K$ **do**
19         Update $\theta, \phi$ with cumulative loss by Adam optimizer:
            $\theta, \phi = \text{argmax}\left(\sum_{n=1}^N L_n(\theta, \phi)\right)$
20     **end**
21     $\theta_{old} \leftarrow \theta$
22 **end**

---

## 2 Details of the Baselines

In this section, we show how the baseline PDRs compute the priority index for the operations. We begin with introducing the notations used in these rules, summarized as follows:

- $Z_{ij}$: the priority index of operation $O_{ij}$;
- $n_i$: the number of operations for job $J_i$;
- $Re_i$: the release time of job $J_i$ (here we assume $Re_i = 0$ for all $J_i$, i.e. all jobs are available in the beginning, but in general the jobs could have different release time);
- $p_{ij}$: the processing time of operation $O_{ij}$.

Based on the above notations, the decision principles for each baseline are given below:

- *Shortest Processing Time* (SPT): min $Z_{ij} = p_{ij}$;
- *Most Work Remaining* (MWKR): max $Z_{ij} = \sum_j^{n_i} p_{ij}$;
- *Minimum ratio of Flow Due Date to Most Work Remaining* (FDD/MWKR): min $Z_{ij} = \left( Re_i + \sum_1^j p_{ij} \right) / \sum_j^{n_i} p_{ij}$;
- *Most Operations Remaining* (MOPNR): max $Z_{ij} = n_i - j + 1$.

## 3 Result on Taillard's Benchmark

Here we present the complete results on Taillard's benchmark. In Table S.1, we report the results of training and testing on 5 groups of instances with sizes up to $30 \times 20$. As we can observe from this table, the PDRs trained by our method outperform the baselines on 92% of these instances (46 out of 50). In Table S.2, we report the generalization performance of our polices trained on $20 \times 20$ and $30 \times 20$ instances. Without training, the two trained PDRs achieve the best performance on all the 30 instances, with the $30 \times 20$ policy performing slightly better.

| | Instance | SPT | MWKR | FDD/WKR | MOPNR | Ours | UB |
|---|---|---|---|---|---|---|---|
| $15 \times 15$ | Ta01 | 1872 (52.1%) | 1786 (45.1%) | 1841 (49.6%) | 1864 (51.4%) | **1443 (17.2%)** | 1231* |
| | Ta02 | 1709 (37.4%) | 1944 (56.3%) | 1895 (52.3%) | 1680 (35.0%) | **1544 (24.1%)** | 1244* |
| | Ta03 | 2009 (64.9%) | 1947 (59.9%) | 1914 (57.1%) | 1558 (27.9%) | **1440 (18.2%)** | 1218* |
| | Ta04 | 1825 (55.3%) | 1694 (44.2%) | 1653 (40.7%) | 1755 (49.4%) | **1637 (39.3%)** | 1175* |
| | Ta05 | 2044 (67.0%) | 1892 (54.6%) | 1787 (46.0%) | **1605 (31.1%)** | 1619 (32.3%) | 1224* |
| | Ta06 | 1771 (43.1%) | 1976 (59.6%) | 1748 (41.2%) | 1815 (46.6%) | **1601 (29.3%)** | 1238* |
| | Ta07 | 2016 (64.3%) | 1961 (59.8%) | 1660 (35.3%) | 1884 (53.5%) | **1568 (27.8%)** | 1227* |
| | Ta08 | 1654 (35.9%) | 1803 (48.2%) | 1803 (48.2%) | 1839 (51.1%) | **1468 (20.6%)** | 1217* |
| | Ta09 | 1962 (54.0%) | 2215 (73.9%) | 1848 (45.1%) | 2002 (57.1%) | **1627 (27.7%)** | 1274* |
| | Ta10 | 2164 (74.4%) | 2057 (65.8%) | 1937 (56.1%) | 1821 (46.7%) | **1527 (23.0%)** | 1241* |
| $20 \times 15$ | Ta11 | 2212 (63.0%) | 2117 (56.0%) | 2101 (54.8%) | 2030 (49.6%) | **1794 (32.2%)** | 1357* |
| | Ta12 | 2414 (76.6%) | 2213 (61.9%) | 2034 (48.8%) | 2117 (54.9%) | **1805 (32.0%)** | 1367* |
| | Ta13 | 2346 (74.7%) | 2026 (50.9%) | 2141 (59.4%) | 1979 (47.4%) | **1932 (43.9%)** | 1343* |
| | Ta14 | 2109 (56.8%) | 2164 (60.9%) | 1841 (36.9%) | 2036 (51.4%) | **1664 (23.7%)** | 1345* |
| | Ta15 | 2163 (61.5%) | 2180 (62.8%) | 2187 (63.3%) | 1939 (44.8%) | **1730 (29.2%)** | 1339* |
| | Ta16 | 2232 (64.1%) | 2528 (85.9%) | 1926 (41.6%) | 1980 (45.6%) | **1710 (25.7%)** | 1360* |
| | Ta17 | 2185 (49.5%) | 2015 (37.8%) | 2093 (43.2%) | 2211 (51.2%) | **1897 (29.8%)** | 1462* |
| | Ta18 | 2267 (62.4%) | 2275 (63.0%) | 2064 (47.9%) | 1981 (41.9%) | **1794 (28.5%)** | 1396 |
| | Ta19 | 2238 (68.0%) | 2201 (65.2%) | 1958 (47.0%) | 1899 (42.6%) | **1682 (26.3%)** | 1332* |
| | Ta20 | 2370 (75.8%) | 2188 (62.3%) | 2195 (62.8%) | 1986 (47.3%) | **1739 (29.0%)** | 1348* |
| $20 \times 20$ | Ta21 | 2836 (72.7%) | 2622 (59.7%) | 2455 (49.5%) | 2320 (41.3%) | **2252 (37.1%)** | 1642* |
| | Ta22 | 2672 (67.0%) | 2554 (59.6%) | 2177 (36.1%) | 2415 (50.9%) | **2102 (31.4%)** | 1600 |
| | Ta23 | 2397 (53.9%) | 2408 (54.7%) | 2514 (61.5%) | 2194 (40.9%) | **2085 (33.9%)** | 1557 |
| | Ta24 | 2787 (69.5%) | 2553 (55.3%) | 2391 (45.4%) | 2250 (36.9%) | **2200 (33.8%)** | 1644* |
| | Ta25 | 2513 (57.6%) | 2582 (61.9%) | 2267 (42.1%) | **2146 (34.5%)** | 2201 (38.0%) | 1595 |
| | Ta26 | 2649 (61.2%) | 2506 (52.5%) | 2644 (60.9%) | 2480 (50.9%) | **2176 (32.4%)** | 1643 |
| | Ta27 | 2707 (61.1%) | 2768 (64.8%) | 2514 (49.6%) | 2298 (36.8%) | **2132 (26.9%)** | 1680 |
| | Ta28 | 2654 (65.6%) | 2370 (47.8%) | 2330 (45.4%) | 2259 (40.9%) | **2146 (33.9%)** | 1603* |
| | Ta29 | 2681 (65.0%) | 2399 (47.6%) | 2232 (37.4%) | 2367 (45.7%) | **1952 (20.1%)** | 1625 |
| | Ta30 | 2662 (68.1%) | 2424 (53.0%) | 2348 (48.2%) | 2370 (49.6%) | **2035 (28.5%)** | 1584 |

| Instance | | SPT | MWKR | FDD/WKR | MOPNR | Ours (20 × 20) | Ours (30 × 20) | UB |
|---|---|---|---|---|---|---|---|---|
| 30 × 15 | Ta31 | 2870 (62.7%) | 2590 (46.8%) | **2459 (39.4%)** | 2576 (46.0%) | 2565 (45.4%) | | 1764* |
| | Ta32 | 3097 (73.6%) | 2725 (52.7%) | 2672 (49.8%) | 2830 (58.6%) | **2388 (33.9%)** | | 1784 |
| | Ta33 | 2782 (55.3%) | 2919 (63.0%) | 2766 (54.4%) | 2746 (53.3%) | **2324 (29.8%)** | | 1791 |
| | Ta34 | 2956 (61.7%) | 2826 (54.6%) | 2669 (46.0%) | 2464 (34.8%) | **2332 (27.6%)** | | 1828* |
| | Ta35 | 2940 (46.5%) | 2791 (39.1%) | 2525 (25.8%) | 2649 (32.0%) | **2505 (24.8%)** | | 2007* |
| | Ta36 | 2933 (61.2%) | 2811 (54.5%) | 2690 (47.9%) | 2666 (46.6%) | **2497 (37.3%)** | | 1819* |
| | Ta37 | 3065 (73.1%) | 2719 (53.5%) | 2492 (40.7%) | 2584 (45.9%) | **2325 (31.3%)** | | 1771* |
| | Ta38 | 2700 (61.4%) | 2706 (61.7%) | 2425 (44.9%) | 2657 (58.8%) | **2302 (37.6%)** | | 1673* |
| | Ta39 | 2698 (50.3%) | 2592 (44.4%) | 2596 (44.6%) | **2409 (34.2%)** | 2410 (34.3%) | | 1795* |
| | Ta40 | 2843 (70.3%) | 2601 (55.8%) | 2614 (56.6%) | 2432 (45.7%) | **2140 (28.2%)** | | 1669 |
| 30 × 20 | Ta41 | 3067 (53.0%) | 3145 (56.9%) | 2991 (49.2%) | 2996 (49.4%) | **2667 (33.0%)** | | 2005 |
| | Ta42 | 3640 (87.9%) | 3394 (75.2%) | 3027 (56.3%) | 2995 (54.6%) | **2664 (37.5%)** | | 1937 |
| | Ta43 | 2843 (54.0%) | 3162 (71.3%) | 2926 (58.5%) | 2666 (44.4%) | **2431 (31.7%)** | | 1846 |
| | Ta44 | 3281 (65.8%) | 3388 (71.2%) | 3462 (74.9%) | 2845 (43.8%) | **2714 (37.1%)** | | 1979 |
| | Ta45 | 3238 (61.9%) | 3390 (69.5%) | 3245 (62.3%) | 3134 (56.7%) | **2637 (31.9%)** | | 2000 |
| | Ta46 | 3352 (67.1%) | 3268 (62.9%) | 3008 (50.0%) | 2802 (39.7%) | **2776 (38.4%)** | | 2006 |
| | Ta47 | 3197 (69.2%) | 2986 (58.1%) | 2940 (55.6%) | 2788 (47.6%) | **2476 (31.1%)** | | 1889 |
| | Ta48 | 3445 (77.9%) | 3050 (57.5%) | 2991 (54.4%) | 2822 (45.7%) | **2490 (28.5%)** | | 1937 |
| | Ta49 | 3201 (63.2%) | 3172 (61.8%) | 2865 (46.1%) | 2933 (49.6%) | **2556 (30.3%)** | | 1961 |
| | Ta50 | 3083 (60.3%) | 2978 (54.9%) | 2995 (55.7%) | 2900 (50.8%) | **2628 (36.7%)** | | 1923 |

Table S.1. **Results on Taillard's Benchmark (Part I).** The "UB" column is the best solution from literature, and "*" means the solution is optimal.

| Instance | | SPT | MWKR | FDD/WKR | MOPNR | Ours (20 × 20) | Ours (30 × 20) | UB |
|---|---|---|---|---|---|---|---|---|
| 50 × 15 | Ta01 | 4280 (55.1%) | 3899 (41.3%) | 3851 (39.5%) | 3616 (31.0%) | 3793 (37.4%) | **3599 (30.4%)** | 2760* |
| | Ta02 | 4419 (60.3%) | 3763 (36.5%) | 3734 (35.5%) | 3698 (34.2%) | 3487 (26.5%) | **3341 (21.2%)** | 2756* |
| | Ta03 | 3949 (45.3%) | 3894 (43.3%) | 3394 (24.9%) | 3402 (25.2%) | **3106 (14.3%)** | 3186 (17.3%) | 2717* |
| | Ta04 | 3977 (40.1%) | 3739 (31.7%) | 3603 (26.9%) | 3599 (26.8%) | 3322 (17.0%) | **3266 (15.0%)** | 2839* |
| | Ta05 | 4307 (60.8%) | 3782 (41.2%) | 3664 (36.8%) | 3650 (36.2%) | 3336 (24.5%) | **3232 (20.6%)** | 2679* |
| | Ta06 | 4156 (49.4%) | 3951 (42.1%) | 4016 (44.4%) | 3638 (30.8%) | 3501 (25.9%) | **3378 (21.5%)** | 2781* |
| | Ta07 | 4321 (46.8%) | 3883 (31.9%) | 3720 (26.4%) | 3705 (25.9%) | 3581 (21.7%) | **3471 (17.9%)** | 2943* |
| | Ta08 | 4090 (41.8%) | 4476 (55.1%) | 3926 (36.1%) | 3661 (26.9%) | **3454 (19.7%)** | 3732 (29.4%) | 2885* |
| | Ta09 | 4101 (54.5%) | 3751 (41.3%) | 3672 (38.3%) | 3530 (33.0%) | 3441 (29.6%) | **3381 (27.3%)** | 2655* |
| | Ta10 | 4347 (59.6%) | 3940 (44.7%) | 3783 (38.9%) | 3581 (31.5%) | **3281 (20.5%)** | 3352 (23.1%) | 2723* |
| 50 × 20 | Ta11 | 4687 (63.4%) | 4313 (50.4%) | 4142 (44.4%) | 3941 (37.4%) | 3830 (33.5%) | **3654 (27.4%)** | 2868* |
| | Ta12 | 4670 (62.8%) | 4542 (58.3%) | 3897 (35.8%) | 4025 (40.3%) | **3617 (26.1%)** | 3722 (29.7%) | 2869* |
| | Ta13 | 4415 (60.3%) | 4069 (47.7%) | 3852 (39.8%) | 3692 (34.0%) | **3397 (23.3%)** | 3536 (28.3%) | 2755* |
| | Ta14 | 4334 (60.4%) | 4176 (54.6%) | 4001 (48.1%) | 3748 (38.7%) | **3275 (21.2%)** | 3631 (34.4%) | 2702* |
| | Ta15 | 4221 (54.9%) | 4600 (68.8%) | 4062 (49.1%) | 3866 (41.9%) | 3510 (28.8%) | **3359 (23.3%)** | 2725* |
| | Ta16 | 4457 (56.7%) | 4209 (47.9%) | 3940 (38.5%) | 3846 (35.2%) | **3388 (19.1%)** | 3555 (25.0%) | 2845* |
| | Ta17 | 4420 (56.5%) | 4172 (47.7%) | 3974 (40.7%) | 3795 (34.3%) | 3848 (36.2%) | **3567 (26.3%)** | 2825* |
| | Ta18 | 4807 (72.7%) | 4428 (59.1%) | 3857 (38.5%) | 4077 (46.4%) | **3514 (26.2%)** | 3680 (32.2%) | 2784* |
| | Ta19 | 4379 (42.6%) | 4758 (54.9%) | 4349 (41.6%) | 4135 (34.6%) | 3763 (22.5%) | **3592 (17.0%)** | 3071* |
| | Ta20 | 4932 (64.7%) | 4484 (49.7%) | 4147 (38.5%) | 4075 (36.1%) | 3976 (32.8%) | **3643 (21.6%)** | 2995* |
| 100 × 20 | Ta21 | 7841 (43.5%) | 6943 (27.1%) | 6818 (24.8%) | 6601 (20.8%) | 6549 (19.9%) | **6452 (18.1%)** | 5464* |
| | Ta22 | 7655 (47.8%) | 7021 (35.5%) | 6358 (22.7%) | 6191 (19.5%) | 5884 (13.6%) | **5695 (9.9%)** | 5181* |
| | Ta23 | 7510 (34.9%) | 7381 (32.6%) | 6967 (25.1%) | 6758 (21.4%) | **6411 (15.1%)** | 6462 (16.1%) | 5568* |
| | Ta24 | 7451 (39.6%) | 6995 (31.0%) | 6381 (19.5%) | 6090 (14.1%) | 5917 (10.8%) | **5885 (10.2%)** | 5339* |
| | Ta25 | 7360 (36.5%) | 7366 (36.6%) | 6757 (25.3%) | 6611 (22.6%) | 6669 (23.7%) | **6355 (17.9%)** | 5392* |
| | Ta26 | 7909 (48.1%) | 7026 (31.5%) | 6641 (24.3%) | 6554 (22.7%) | 6337 (18.6%) | **6135 (14.8%)** | 5342* |
| | Ta27 | 7456 (37.2%) | 7502 (38.0%) | 6540 (20.3%) | 6589 (21.2%) | 6297 (15.8%) | **6056 (11.4%)** | 5436* |
| | Ta28 | 7400 (37.2%) | 6861 (27.2%) | 6750 (25.1%) | 6313 (17.0%) | 6177 (14.5%) | **6101 (13.1%)** | 5394* |
| | Ta29 | 7743 (44.5%) | 7232 (35.0%) | 6461 (20.6%) | 6665 (24.4%) | 6185 (15.4%) | **5943 (10.9%)** | 5358* |
| | Ta30 | 7321 (41.3%) | 6961 (34.3%) | 6534 (26.1%) | 6151 (18.7%) | 6124 (18.2%) | **5892 (13.7%)** | 5183* |

Table S.2. **Results on Taillard's Benchmark (Part II).** The "UB" column is the best solution from literature, and "*" means the solution is optimal.

## 4 Result on DMU Benchmark

Similar conclusion can be drawn from results on DMU benchmark. In Table S.3, we report results of training and testing on 4 groups of instances with sizes up to 30 × 20, where our method outperforms baselines over 87.5% (35 out of 40) of these instances. In Table S.4 which focuses on the generalization performance, our policies trained on 20 × 20 and 30 × 20 instances outperform the baselines on 77.5% (31 out of 40) instances.

| | Instance | SPT | MWKR | FDD/WKR | MOPNR | Ours | UB |
|---|---|---|---|---|---|---|---|
| 20 × 15 | Dmu01 | 4516 (76.2%) | 3988 (55.6%) | 3535 (37.9%) | 3882 (51.5%) | **3323 (29.7%)** | 2563 |
| | Dmu02 | 4593 (69.7%) | 4555 (68.3%) | 3847 (42.2%) | 3884 (43.5%) | **3630 (34.1%)** | 2706 |
| | Dmu03 | 4438 (62.5%) | 4117 (50.8%) | 4063 (48.8%) | 3979 (45.7%) | **3660 (34.0%)** | 2731* |
| | Dmu04 | 4533 (69.8%) | 3995 (49.7%) | 4160 (55.9%) | 4079 (52.8%) | **3816 (43.0%)** | 2669 |
| | Dmu05 | 4420 (60.8%) | 4977 (81.0%) | 4238 (54.2%) | 4116 (49.7%) | **3897 (41.8%)** | 2749* |
| | Dmu41 | 5283 (62.7%) | 5377 (65.5%) | 5187 (59.7%) | 5070 (56.1%) | **4316 (32.9%)** | 3248 |
| | Dmu42 | 5354 (57.9%) | 6076 (79.2%) | 5583 (64.7%) | 4976 (46.8%) | **4858 (43.3%)** | 3390 |
| | Dmu43 | 5328 (54.8%) | 4938 (43.5%) | 5086 (47.8%) | 5012 (45.7%) | **4887 (42.0%)** | 3441 |
| | Dmu44 | 5745 (64.7%) | 5630 (61.4%) | 5550 (59.1%) | 5213 (49.5%) | **5151 (47.7%)** | 3488 |
| | Dmu45 | 5305 (62.1%) | 5446 (66.4%) | 5414 (65.5%) | 4921 (50.4%) | **4615 (41.0%)** | 3272 |
| 20 × 20 | Dmu06 | 6230 (92.0%) | 5556 (71.3%) | 5258 (62.1%) | 4747 (46.3%) | **4358 (34.3%)** | 3244 |
| | Dmu07 | 5619 (84.5%) | 4636 (52.2%) | 4789 (57.2%) | 4367 (43.4%) | **3671 (20.5%)** | 3046 |
| | Dmu08 | 5239 (64.3%) | 5078 (59.3%) | 4817 (51.1%) | 4480 (40.5%) | **4048 (27.0%)** | 3188 |
| | Dmu09 | 4874 (57.6%) | 4519 (46.2%) | 4675 (51.2%) | 4519 (46.2%) | **4482 (45.0%)** | 3092 |
| | Dmu10 | 4808 (61.1%) | 4963 (66.3%) | 4149 (39.0%) | 4133 (38.5%) | **4021 (34.8%)** | 2984 |
| | Dmu46 | 6403 (58.7%) | 6168 (52.9%) | **5778 (43.2%)** | 6136 (52.1%) | 5876 (45.6%) | 4035 |
| | Dmu47 | 6015 (52.7%) | 6130 (55.6%) | 6058 (53.8%) | 5908 (50.0%) | **5771 (46.5%)** | 3939 |
| | Dmu48 | 5345 (42.0%) | 5701 (51.5%) | 5887 (56.4%) | 5384 (43.1%) | **5034 (33.8%)** | 3763 |
| | Dmu49 | 6072 (63.7%) | 6089 (64.1%) | 5807 (56.5%) | **5469 (47.4%)** | 5470 (47.4%) | 3710 |
| | Dmu50 | 6300 (68.9%) | 6050 (62.2%) | 5764 (54.6%) | 5380 (44.3%) | **5314 (42.5%)** | 3729 |
| 30 × 15 | Dmu11 | 5864 (71.0%) | 4961 (44.6%) | 4798 (39.9%) | 4891 (42.6%) | **4435 (29.3%)** | 3430 |
| | Dmu12 | 5966 (70.7%) | 5994 (71.5%) | 5595 (60.1%) | 4947 (41.5%) | **4864 (39.2%)** | 3495 |
| | Dmu13 | 5744 (56.0%) | 6190 (68.2%) | 5324 (44.6%) | 4979 (35.3%) | **4918 (33.6%)** | 3681* |
| | Dmu14 | 5469 (61.1%) | 5567 (64.0%) | 4830 (42.3%) | 4839 (42.6%) | **4130 (21.7%)** | 3394* |
| | Dmu15 | 5518 (65.1%) | 5299 (58.5%) | 4928 (47.4%) | 4653 (39.2%) | **4392 (31.4%)** | 3343* |
| | Dmu51 | 6538 (56.9%) | 6841 (64.2%) | 7002 (68.0%) | 6691 (60.6%) | **6241 (49.8%)** | 4167 |
| | Dmu52 | 7341 (70.3%) | 6942 (61.0%) | 6650 (54.3%) | **6591 (52.9%)** | 6714 (55.7%) | 4311 |
| | Dmu53 | 7232 (64.6%) | 7430 (69.1%) | 7170 (63.2%) | 6851 (55.9%) | **6724 (53.0%)** | 4394 |
| | Dmu54 | 7178 (64.6%) | **6461 (48.1%)** | 6767 (55.1%) | 6540 (49.9%) | 6522 (49.5%) | 4362 |
| | Dmu55 | **6212 (45.4%)** | 6844 (60.2%) | 7101 (66.3%) | 6446 (50.9%) | 6639 (55.4%) | 4271 |
| 30 × 20 | Dmu16 | 6241 (66.4%) | 5837 (55.6%) | 5948 (58.6%) | 5743 (53.1%) | **4953 (32.0%)** | 3751 |
| | Dmu17 | 6487 (70.1%) | 6610 (73.3%) | 6035 (58.2%) | 5540 (45.3%) | **5379 (41.0%)** | 3814 |
| | Dmu18 | 6978 (81.5%) | 6363 (65.5%) | 5863 (52.5%) | 5714 (48.6%) | **5100 (32.7%)** | 3844* |
| | Dmu19 | 5767 (53.1%) | 6385 (69.5%) | 5424 (43.9%) | 5223 (38.6%) | **4889 (29.8%)** | 3768 |
| | Dmu20 | 6910 (86.3%) | 6472 (74.4%) | 6444 (73.7%) | 5530 (49.1%) | **4859 (31.0%)** | 3710 |
| | Dmu56 | 7698 (55.8%) | 7930 (60.5%) | 8248 (66.9%) | 7620 (54.2%) | **7328 (48.3%)** | 4941 |
| | Dmu57 | 7746 (66.4%) | 7063 (51.7%) | 7694 (65.3%) | 7345 (57.8%) | **6704 (44.0%)** | 4655 |
| | Dmu58 | 7269 (54.4%) | 7708 (63.7%) | 7601 (61.4%) | 7216 (53.3%) | **6721 (42.8%)** | 4708 |
| | Dmu59 | 7114 (53.8%) | 7335 (58.6%) | 7490 (62.0%) | 7589 (64.1%) | **7109 (53.7%)** | 4624 |
| | Dmu60 | 8150 (71.4%) | 7547 (58.7%) | 7526 (58.3%) | 7399 (55.6%) | **6632 (39.5%)** | 4755 |

Table S.3. **Results on DMU Benchmark (Part I).** The "UB" column is the best solution from literature, and "*" means the solution is optimal.

| | Instance | SPT | MWKR | FDD/WKR | MOPNR | Ours (20 × 20) | Ours (30 × 20) | UB |
|---|---|---|---|---|---|---|---|---|
| 40 × 15 | Dmu21 | 7400 (68.9%) | 6314 (44.2%) | 6416 (46.5%) | 6048 (38.1%) | 5559 (26.9%) | **5317 (21.4%)** | 4380* |
| | Dmu22 | 7353 (55.6%) | 6980 (47.7%) | 6645 (40.6%) | 6351 (34.4%) | 5929 (25.5%) | **5534 (17.1%)** | 4725* |
| | Dmu23 | 7262 (55.6%) | 6472 (38.6%) | 6781 (45.3%) | 6004 (28.6%) | 5681 (21.7%) | **5620 (20.4%)** | 4668* |
| | Dmu24 | 6799 (46.3%) | 7079 (52.3%) | 6582 (41.6%) | 6155 (32.4%) | **5479 (17.9%)** | 5753 (23.8%) | 4648* |
| | Dmu25 | 6428 (54.4%) | 6042 (45.1%) | 5756 (38.2%) | 5365 (28.8%) | 4825 (15.9%) | **4775 (14.7%)** | 4164* |
| | Dmu61 | **7817 (51.1%)** | 8734 (68.9%) | 8757 (69.3%) | 8076 (56.1%) | 8053 (55.7%) | 8203 (58.6%) | 5172 |
| | Dmu62 | **7759 (47.4%)** | 8262 (56.9%) | 8082 (53.5%) | 8253 (56.8%) | 8415 (59.8%) | 8091 (53.7%) | 5265 |
| | Dmu63 | 8296 (55.8%) | 8364 (57.0%) | 8384 (57.4%) | 8417 (58.0%) | 8330 (56.4%) | **8031 (50.8%)** | 5326 |
| | Dmu64 | 8444 (60.8%) | 8406 (60.1%) | 8490 (61.7%) | 8161 (55.4%) | 7916 (50.8%) | **7738 (47.4%)** | 5250 |
| | Dmu65 | 8454 (62.9%) | 8189 (57.8%) | 8307 (60.1%) | 8225 (58.5%) | 8093 (55.9%) | **7577 (46.0%)** | 5190 |
| 40 × 20 | Dmu26 | 7766 (67.1%) | 7107 (52.9%) | 7240 (55.8%) | 6236 (34.2%) | **5908 (27.1%)** | 5946 (28.0%) | 4647* |
| | Dmu27 | 7501 (54.7%) | 7313 (50.8%) | 6965 (43.7%) | 6936 (43.1%) | 6542 (34.9%) | **6418 (32.4%)** | 4848* |
| | Dmu28 | 8621 (83.7%) | 8194 (74.6%) | 6516 (38.9%) | 6714 (43.1%) | 6272 (33.7%) | **5986 (27.6%)** | 4692* |
| | Dmu29 | 8052 (71.6%) | 7448 (58.8%) | 6971 (48.6%) | 6990 (49.0%) | 6169 (31.5%) | **6051 (29.0%)** | 4691* |
| | Dmu30 | 7372 (55.8%) | 7890 (66.7%) | 6910 (46.0%) | 6869 (45.2%) | 6022 (27.3%) | **5988 (26.5%)** | 4732* |
| | Dmu66 | 8971 (56.9%) | 8966 (56.8%) | 9606 (68.0%) | 8726 (52.6%) | 8547 (49.5%) | **8475 (48.2%)** | 5717 |
| | Dmu67 | 9096 (56.5%) | 9306 (60.1%) | 9103 (56.6%) | 9372 (61.2%) | **8791 (51.2%)** | 8832 (51.9%) | 5813 |
| | Dmu68 | 9265 (60.5%) | 9445 (63.6%) | 9431 (63.4%) | 8722 (51.1%) | 9117 (57.9%) | **8693 (50.6%)** | 5773 |
| | Dmu69 | 9215 (61.4%) | 9450 (65.5%) | 9951 (74.3%) | 8697 (52.3%) | 9130 (59.9%) | **8634 (51.2%)** | 5709 |
| | Dmu70 | 9522 (61.7%) | 9490 (61.1%) | 9416 (59.9%) | 9445 (60.4%) | **8601 (46.1%)** | 8735 (48.3%) | 5889 |

| | | | | | | | UB |
|---|---|---|---|---|---|---|---|
| 50 × 15 | Dmu31 | 8869 (57.3%) | 8147 (44.5%) | 7899 (40.1%) | 7192 (27.5%) | 7191 (27.5%) | **7156 (26.9%)** | 5640* |
| | Dmu32 | 7814 (31.8%) | 8004 (35.0%) | 7316 (23.4%) | 7267 (22.6%) | 6938 (17.1%) | **6506 (9.8%)** | 5927* |
| | Dmu33 | 8114 (41.7%) | 7710 (34.6%) | 7262 (26.8%) | 7069 (23.4%) | 6480 (13.1%) | **6192 (8.1%)** | 5728* |
| | Dmu34 | 7625 (41.6%) | 7709 (43.2%) | 7725 (43.5%) | 6919 (28.5%) | 6661 (23.7%) | **6257 (16.2%)** | 5385* |
| | Dmu35 | 8626 (53.1%) | 7617 (35.2%) | 7099 (26.0%) | 7033 (24.8%) | 6417 (13.9%) | **6302 (11.8%)** | 5635* |
| | Dmu71 | 9594 (53.9%) | 9978 (60.1%) | 10889 (74.7%) | **9514 (52.6%)** | 9950 (59.6%) | 9797 (57.2%) | 6233 |
| | Dmu72 | **9882 (52.4%)** | 10135 (56.3%) | 11602 (79.0%) | 10063 (55.2%) | 10401 (60.4%) | 9926 (53.1%) | 6483 |
| | Dmu73 | 9953 (61.5%) | 9721 (57.7%) | 10212 (65.7%) | **9615 (56.0%)** | 10080 (63.6%) | 9933 (61.2%) | 6163 |
| | Dmu74 | 9866 (58.6%) | 10086 (62.2%) | 10659 (71.4%) | **9536 (53.3%)** | 10445 (67.9%) | 9833 (58.1%) | 6220 |
| | Dmu75 | **9411 (51.9%)** | 9953 (60.6%) | 10839 (74.9%) | 10157 (63.9%) | 9937 (60.4%) | 9892 (59.6%) | 6197 |
| 50 × 20 | Dmu36 | 9911 (76.3%) | 8090 (43.9%) | 8084 (43.8%) | 7703 (37.0%) | **7213 (28.3%)** | 7470 (32.9%) | 5621* |
| | Dmu37 | 8917 (52.4%) | 9685 (65.5%) | 9433 (61.2%) | 7844 (34.1%) | 7765 (32.7%) | **7296 (24.7%)** | 5851* |
| | Dmu38 | 9384 (64.3%) | 8414 (47.3%) | 8428 (47.5%) | 8398 (47.0%) | 7429 (30.0%) | **7410 (29.7%)** | 5713* |
| | Dmu39 | 9221 (60.4%) | 9266 (61.2%) | 8177 (42.3%) | 7969 (38.7%) | 7168 (24.7%) | **6827 (18.8%)** | 5747* |
| | Dmu40 | 9406 (68.7%) | 8261 (48.1%) | 7773 (39.4%) | 8173 (46.5%) | 7757 (39.1%) | **7325 (31.3%)** | 5577* |
| | Dmu76 | 11677 (71.4%) | 10571 (55.2%) | 11576 (69.9%) | 11019 (61.7%) | 10322 (51.5%) | **9698 (42.3%)** | 6813 |
| | Dmu77 | **10401 (52.5%)** | 11148 (63.4%) | 11910 (74.6%) | 10577 (55.0%) | 10729 (57.3%) | 10693 (56.7%) | 6822 |
| | Dmu78 | 10585 (56.4%) | 10540 (55.7%) | 11464 (69.3%) | 10989 (62.3%) | 10742 (58.7%) | **9986 (47.5%)** | 6770 |
| | Dmu79 | 11115 (59.5%) | 11201 (60.7%) | 11035 (58.3%) | **10729 (53.9%)** | 10993 (57.7%) | 10936 (56.9%) | 6970 |
| | Dmu80 | 10711 (60.2%) | 10894 (62.9%) | 11116 (66.3%) | 10679 (59.7%) | 10041 (50.2%) | **9875 (47.7%)** | 6686 |

Table S.4. **Results on DMU Benchmark (Part II).** The "UB" column is the best solution from literature, and "*" means the solution is optimal.

# 5  Training Curve

We show training curves for all problems in Figure.1. The problem sizes are $\{6 \times 6, 10 \times 10, 15 \times 15, 20 \times 15, 20 \times 20, 30 \times 15, 30 \times 20\}$ respectively. In each curve, after learning on every 200 totally new instances, the averaged performance (makespan) over these 200 instances is plotted. The training time for each problem is: 0.95h ( for 1a), 2.6h (for 1b), 6.2h (for 1c), 11.6h (for 1d), 20.3h (for 1e), 8.7h (1f), 11.6h (1g), 13.5h (1h), and 20.3h (1i) respectively.

Figure 1: **Training curves for all problems.** The scale of processing time of each problem is given in braces, e.g. $\{1, 99\}$ indicating the scale of processing time is a integer uniformly distributed in range from 1 to 99. Sizes $20 \times 20$ and $30 \times 20$ have 2 different scales $\{1, 99\}$ and $\{1, 199\}$ respectively.