[Reviews · NeurIPS 2020]

Review 1

Summary and Contributions: The authors propose an RL framework to produce a simple and effective priority dispatch rule (PDR) for job-shop scheduling problem. Job-shop scheduling are difficult optimization problems and very sofisticate methods following both integer programming and constraint programming paradigms have been devised. PDRs are generally way simpler than that: essentially MDPs ordering decisions according to a rule and producing a solution accordingly. The proposal is to use RL to learn a more sofisticate PDR still at a reasonable short computing cost (at test time).

Strengths: The paper is essentially strong on the computational side. The results are very good wrt PDRs, beating classical static rules both on synthetic instances and on ones from the literature.

Weaknesses: The level of innovation from the methodology standpoint does not look high and the computational evaluation does not say how good is the devised heuristic wrt good scheduling heuristics. ========================== AFTER REBUTTAL ===================== I appreciate the answers of the authors and their effort to clarify their contribution. It remains a bit unclear to me if beating PDRs is as significant as the authors claim but overall I kept my positive score.

Correctness: The paper is correct as far as I can see.

Clarity: The paper is well written although I would have appreciated that Figure 1 depicting an example of job-shop and solution of it were discussed a bit more in details including numerical indication of the start times.

Relation to Prior Work: The previous work seems to be covered correctly.

Reproducibility: Yes

Additional Feedback: As mentioned, to be totally convincing from the computational standpoint, one needs to know a comparison with more sophisticated heuristics for the same problem. In addition, some job-shop problems can have feasibility constraints, for example due dates. How does the approach extend to such a situation.


Review 2

Summary and Contributions: The paper proposes to learn priority dispatching rules (PDR) for job shop problems using graph neural networks and training via reinforcement learning. The job shop instance is formulated as a disjunctive graph and the RL agent chooses which nodes to dispatch, i.e. which task should be performed next. Experiments are performed on generated and known instances and the method is compared to several PDRs from the literature and an exact CP solver. The result show to be competitive with the baseline methods on quality but not always faster.

Strengths: The paper does a good job to explain the proposed method and justify them using the graph representation of the job shop scheduling problem. The method is based on a size-invariant graph neural network representation, which is currently the most favored network model for these kinds of problems.

Weaknesses: The focus on PDRs for job shop limits the further generalization of the method to other problems whereas a recent direction in research seems to be to learn general methods for combinatorial optimization using DL/RL. However, there is also interest to find methods for specific applications and this seems to be a good example for one.

Correctness: The described method appears correct and represents formulations and representations from the existing literature where applicable.

Clarity: The paper is well written.

Relation to Prior Work: Related work is sufficiently discussed.

Reproducibility: No

Additional Feedback: I find it interesting that we see that the proposed method has better quality, but longer time, whereas we usually observe opposite behavior for combinatorial optimization using DL. Could the authors elaborate on this? Details on the algorithm and its implementation are given and it could be possible to reimplement the method, but it would be preferable to provide executable code for reproducibility. Update after feedback: Thanks to the authors for clarifying my questions on the reason for longer runtimes.


Review 3

Summary and Contributions: This paper proposes a new machine-learning based approach for tackling generic job-shop scheduling problems (JSSP), by introducing a dedicated graph-based neural network model, and a reinforcement learning-based methodology for learning priority dispatching policies. The authors present an empirical evaluation on both synthetic and real-world datasets, where the proposed approach is shown to improve upon a series of state-of-the-art dispatching rules from the literature. --------------------------------------- POST-AUTHOR-RESPONSE --------------------------------------- I have read the author's response letter, and despite the fact that the value of improving upon PDRs is questionable for the JSSP problem, I remain positive about the paper. > The main reason for not testing on larger instances is lacking benchmarking solutions Indeed, you can not report performances as a percentage in that case, but that is not a valid reason for not performing the experiment. Reporting raw objective values is perfectly fine I believe, if you report them for several methods. the results you report in the additional experiments are very convincing, and will add a substantial value to the paper. The exponential solving times for OR-tools on 5x5, 10x10 and 15x15 instances also provide a valuable piece of information to the reader. After you include both those results in the manuscript, and answer the few concerns raised by the reviewers, you'll have a fine paper IMO.

Strengths: I found the paper clear and well-motivated. While the proposed approach is not fundamentally new, i.e., a lot of works have already investigated similar reinforcement learning-based approaches for solving an NP-hard problem, I believe the proposed graph neural-network model is novel, and makes a lot of sense considering the graphical formulation of the job-shop scheduling problem. I also believe the problem tackled in the paper, job-shop scheduling, is relevant, and that the proposed methodology can be of practical importance to practitioners.

Weaknesses: A first limitation of this work, I found, is that the conducted experiments are limited to rather small problem sizes. While I agree that the problem instances presented in the paper can be extremely difficult to solve exactly, I also believe that the approximate methods used as baselines, as well as the proposed GNN method, can easily be applied to much larger problems. I would be very curious to see how those would behave on such large problems. A second limitation would be that the paper is missing information regarding the cost of the proposed method in training. How long does it take to train the model ? Does the model always converge ?

Correctness: While I have minor comments which I believe the authors can and should address, I did not find any major methodological flaw in the paper.

Clarity: I found the paper very well written, and easy to read. The authors also provide a lot of implementation details (learning rate, hidden dimensions etc.), and plan to provide the source code, therefore I believe the paper is reproducible.

Relation to Prior Work: The paper adequately relates to previous works on reinforcement learning approaches for tackling combinatorial optimization problems.

Reproducibility: Yes

Additional Feedback: Global comments: I think the authors did not use the proper NeurIPS LaTex template. Line numbers are missing, as well as footnotes. The global look and number of pages seem ok though. Figures 1, 2 and 3 are of poor quality, as they appear visually pixelated. I strongly suggest the authors to use images in vector graphics. Detailled comments p.1 paragraph 2: existing methods cannot apply due to the structural differences -> I do not understand that sentence p.2 paragraph 6: each element O_ij [...] -> Which value does O_ij take ? Is O_ij a machine id ? p.2 paragraph 6: p_ij \in N -> can processing times be nagative ? p.3 figure 2: all disjunctive arcs are assigned with directions -> This is misleading. In the picture not all disjunctive arcs have been directed. Some have been removed. I see that those can be safely removed, as ancestrality is preserved. However you might want to clarify that in the text, or not remove them in the figure, for the sake of consistency. p.3 paragraph 6: the completion time of O -> what is meant by completion time here ? The processing time ? The job start time + processing time ? p.4 Figure 2: This example is missing the starting time of each sheduled operation. Also, what do orange nodes mean ? Already scheduled operations ? Another missing information here, what is the action space ? {O_13, O_22, O_32} ? Finally, although I tried to guess the answer to those questions, I do not understand this example. If action O_11 has already been selected, then the operation has already been allocated to the earliest feasible time period (according to the text). Therefore it should have been scheduled for time t=0, and there should be two directed arcs O_11 -> O_22 and O_11 -> O_32. How then is it possible to schedule O_22 before O_11 ? p.4 paragraph 7: which is clearly non-euclidian -> What does this mean ? Non-linear ? p.4 paragraph 7: takes input -> takes as input p.6 section 5: Experiments -> This section is missing some information about training times. How long does it take to train the GNN model ? Hours ? Days ? Weeks ? p.7 table 1: I would be curious in knowing the solving time of OR-Tools on the small instances, for comparison. p.7 table 1: instances that OR-Tools returns -> instances for which OR-Tools returns p.7 table 2: Even the largest instances here look pretty small (100 jobs, 20 machines), and the reported running times for the compared baselines are still very reasonable, 30 secs max. What prevents you from running experiments on even larger instances ? Say, 200x50 ? Applying your method to instances an order of magnitude larger than the training ones would be much more convincing, as an argument for generalizability. If it does not to generalize for instances too different from the training ones, then that information is important as well. It is ok to report negative results. p.7 table 2: Generalizaiton restuls -> typo p.7 paragraph 1: our method [...] can solve the largest instance -> this is a big no. Your method does not solve instances. Solving, in the combinatorial optimization community, means providing an optimal solution, along with its proof of optimality. You can only claim that your method solves approximately an instance, or more accurately provides an approximate solution to the instance. p.8 paragraph 2: a MDP -> an MDP


Review 4

Summary and Contributions: This paper proposes to learn priority dispatching rules (PDRs) for solving Job-shop scheduling problems (JSSP) via Proximal Policy Optimization (PPO). By representing the JSSP into disjunctive graphs, the authors develop a novel Graph Neural Network (GNN)-based policy that operates on the graph structure with raw input features and outputs scheduling actions for constructing the final solution, which serves as the main contribution. The proposed method is evaluated both on generated instances and public JSSP benchmarks and they show superior results to other hand-crafted PDRs. ============= After Rebuttall ============== After discussing the paper and rebuttal with the other reviewers, I was left with one primary concern: It is unclear whether this paper is quite yet a contribution to the JSSP community. The architecture is novel, the results promising, and paper well-written. However, the paper did not show that the algorithm forms a point on the Pareto front trading off computation time and solution quality. I encourage the authors to benchmark directly against JSSP meta-heuristics and other GNN architectures and more thoroughly explore the trade-offs, showing that the current approach offers a new SOTA in some respect w.r.t. JSSP literature. For example, see: Zhang, L. and Wong, T.N., 2015. An object-coding genetic algorithm for integrated process planning and scheduling. European Journal of Operational Research, 244(2), pp.434-444.

Strengths: +This paper extends the recent idea of learning effective heuristics for solving combinatorial optimization problems via deep learning. +Compared to other learning-based methods in solving JSSP, this paper’s novelty is the use of GNN to develop a size-agnostic policy that generalizes to unseen larger problems. +By combining Graph Isomorphism Network (GIN) with PPO and adapting them to fit smoothly with the disjunctive graph formulation, the agent can learn policies that show strong performance against other popular PDRs on several datasets. +This paper is also a good effort towards connecting the Operations Research community with deep learning researchers.

Weaknesses: -There is little discussion on the justification for choosing GIN instead of other recent GNN architectures, such as the graph attention network in [12] which has been used in solving routing problems via reinforcement learning. -While the trained model shows better performance over four traditional hand-crafted HDRs, its gap to solutions found by Google OR-Tools is around 20% or higher for most of the generated problems. This gap also exists on public benchmarks between the learned policy and best solutions in literature, thus making the GNN-based policy less competitive or convincing. I would like to see a comparison of the proposed algorithm to an algorithm well-tuned to those training problems, as well as non-HDR-based heuristics. -The authors argue several advantages of the GNN formulation at end of Sec. 4.2. However, they only evaluate the first advantage on scalability. As both the feature and reward are related to makespan and the processing times are deterministic, I’m suspicious that it can be easily extended to deal with uncertainty/randomness and other shop scheduling problems. -The paper is missing some recent publications on the topic of applying GNNs to scheduling problems, though, some of them would be "contemporary" to this work: Lv, M., Hong, Z., Chen, L., Chen, T., Zhu, T. and Ji, S., 2020. Temporal Multi-Graph Convolutional Network for Traffic Flow Prediction. IEEE Transactions on Intelligent Transportation Systems. Sun, P., Guo, Z., Wang, J., Li, J., Lan, J. and Hu, Y., 2020. DeepWeave: Accelerating Job Completion Time with Deep Reinforcement Learning-based Coflow Scheduling. In Proceedings of the Twenty-Ninth International Joint Conference on Artificial Intelligence, IJCAI-20. Wang, Z. and Gombolay, M., 2020. Learning Scheduling Policies for Multi-Robot Coordination With Graph Attention Networks. IEEE Robotics and Automation Letters, 5(3), pp.4509-4516.

Correctness: Yes.

Clarity: Most of the parts are well written. However, the example given in Sec. 4.1 “State transition” is confusing. It is not clear why Q22 can be allocated before Q11. Will this affect the lower bound of the Q11 completion time? It is possible that there is a helper function deciding the actual arc to be added given the selected operation. If so, please provide more details on detail.

Relation to Prior Work: Mostly.

Reproducibility: Yes

Additional Feedback: -Typo in the caption of Table 2. -Why pick actions greedily during testing instead of sticking to the same sampling strategy as used in training? -In “Models and configurations”, how are those hyper-parameters chosen? Is there any cross-validation involved? -The proposed framework is examined under minimizing the total makespan. How does it perform on a different objective function, such as tardiness? -The paper does not provide an ablation study on the GNN design, such as how K and hidden dimensions affect the learning process and final performance.

[Author Response · NeurIPS 2020]

We greatly appreciate the valuable comments from all reviewers.

**General response: comparison with sophisticated heuristics.** Both reviewer #1 and #4 point out the necessity of
comparing with more sophisticated non-PDR heuristics (often complicated metaheuristics). On one hand, it is not quite
fair to compare with them since they often consume much longer time to iteratively explore the solution space, while
PDR heuristic is a one-time quick construction process. On the other hand, though not explicitly mentioned, over 90%
of the best results for the Taillard's and DMU benchmarks (generated using the same procedure we used to sample
training instances) are obtained by well-tuned metaheuristics as listed in the links we provide in the paper. Hence in
Table 3 and 4, we are actually comparing with the best results from a collection of well-tuned, non-PDR heuristics.

**Reviewer#1**
**Regarding strong heuristic.** Please kindly refer to the general response above.
**Regarding feasibility constraints.** In job shop scheduling, due dates are often considered as soft constraints, i.e.
violations incur tardiness with costs. The optimization objective in such case is often in the form of minimizing
weighted tardiness. Nevertheless, for the cases where due dates are hard constraints, our method can also be applied by
introducing extra (large) penalty into the reward such that the agent could learn to avoid violating the hard constraints.

**Reviewer#2**
**Regarding better quality but longer time.** Our baselines are traditional hand-crafted PDRs, which are derived from
human experience and often give solutions relatively far from optimality. The better solution quality of our method
comes from the fact that our PDRs are automatically learned and optimized by the reinforcement learning system, which
exceeds the limitation of human knowledge. On the other hand, traditional PDRs are computationally cheap while our
PDRs need additional overhead in the inference of GNN, hence have longer (but acceptable) computational time.
**Regarding reproducibility.** As we mentioned in the paper, all source code and trained models will be released.

**Reviewer#3**
**Regarding generalization.** The main reason for not testing on larger instances is lacking benchmarking solutions
since OR-Tools already times out on 98% of 100x20 ones. Here we quickly run our 30x20 policy and baselines on 10
200x50 instances generated using Taillard's method. The average makespans are: SPT (15879.8), MWKR (15391.7),
FDD/MWKR (14095.0), MOPNR (13387.3), and Ours (13070.8). This shows that our policy still performs better than
traditional PDRs on instances an order of magnitude larger than those used in training.
**Regarding training cost.** The training time of all sizes is: 0.95h (6x6), 2.6h (10x10), 6.2h (15x15), 8.7h (20x15),
11.6h (20x20), 13.5h (30x15), 20.3h (30x20). Since training is offline, such cost is reasonable considering the good
online quality and speed. Training always converges, and we will present training costs and curves in the final version.
**Regarding OR-Tools time.** The average time of OR-Tools is 0.012s, 0.027s, and 211.814s for size 6x6, 10x10, and
15x15, respectively. For sizes larger than 20x20, OR-Tools are timed out on most instances, i.e. exceeds 3600s.
**Regarding the example in Figure 2.** We thank the reviewer for pointing out this error. In the case of Figure 2, $O_{22}$
will always be a successor of $O_{11}$. We will correct this mistake in the final version.
For other detailed comments, we greatly appreciate the careful review and will incorporate them in the final version.

**Reviewer#4**
**Regarding GIN.** Our main focus is to design a size-agnostic end-to-end model via GNN that can automatically learn
strong PDRs. However, we are not restricted to particular GNN models. We choose GIN mainly because it is one of the
strongest GNN architectures with proved discriminative power. We will try other GNNs in the future.
**Regarding well-tuned heuristic.** Please kindly refer to the general response above.
**Regarding uncertainty and other shop scheduling.** While we focus on deterministic setting here, our method could
potentially handle uncertainty since it is naturally captured by MDP. The feature and reward used here are not necessarily
deterministic, and could be replaced by expectations/observations of random variables. For other shop scheduling
problems, as long as they share the disjunctive graph representation, they can be handled by our method which operates
directly on the disjunctive graphs. As mentioned in the paper, both directions will be investigated in the future.
**Regarding contemporary publications.** We thank the reviewer for pointing out these papers. They are indeed relevant,
though they are not solving JSSP and do not exploit the disjunctive graph representation scheme. We will cite them
properly in the final version.
**Regarding hyper-parameters and GNN design.** The architecture of our model (e.g. K, and hidden dimension) and
all the hyper-parameters are empirically tuned on small instances and fixed for both training and evaluation.
**Regarding other objectives.** Currently we focus on makespan which is the most common objective for JSSP. Never-
theless, our method can be applied to other objectives such as tardiness by modifying the reward function. We will
investigate this in the future.
**Regarding the example in Figure 2.** We thank the reviewer for pointing out this error. In the case of Figure 2, $O_{22}$
will always be a successor of $O_{11}$. We will correct this mistake in the final version.
**Regarding greedy policy during testing.** For testing, greedy policy is much faster for inference and still gives good
results (see [15]). Sampling multiple solutions will certainly yield better results, and we will try it in the future.

[Meta-Review · NeurIPS 2020]

The reviewers all agree that the paper is above the acceptance threshold. There is novelty in how the paper applies learning to JSSP, and the results are promising. But as Reviewer 4 points out, the paper hasn't shown how the proposed approach trades off solution quality and running time, without which it is difficult to judge whether it is a significant advance over existing techniques. Adding such results will strengthen the paper considerably.